# First Report on Mesophyll Protoplast Isolation and Regeneration System for the *Duboisia* Species

**DOI:** 10.3390/plants13010040

**Published:** 2023-12-21

**Authors:** Yuxin Xue, Jayeni Chathurika Amarathunga Hiti-Bandaralage, Zhangpan Hu, Zizhu Zhao, Neena Mitter

**Affiliations:** 1Centre for Horticultural Science, Queensland Alliance for Agriculture and Food Innovation, The University of Queensland, Brisbane, QLD 4068, Australia; yuxin.xue@uq.net.au (Y.X.); magnolia.hu@uq.edu.au (Z.H.); 2School of Agriculture and Food Sciences, The University of Queensland, Brisbane, QLD 4072, Australia; zizhu.zhao@uq.net.au

**Keywords:** *Duboisia* spp., corkwood, medicinal plant, mesophyll protoplast isolation, protoplast yield, plating density, microcalli formation, tissue culture, thidiazuron, regeneration of plants

## Abstract

The *Duboisia* species, a group of plants native to Australia, have been historically valued for their pharmacological properties and have played a significant role in traditional medicine and pharmaceutical research. Persistent efforts are underway to enhance the efficacy of the active ingredient scopolamine, employing both conventional breeding methods and advanced biotechnology tools. The primary objective of this research was to establish a highly efficient method for isolating mesophyll protoplasts and facilitating their regeneration, thereby laying a robust foundation for the application of various advanced plant biotechnology tools in the pursuit of genetic enhancement. The mesophyll protoplast isolation process was developed for hybrid *D. myoporoides* × *D. hopwoodii* with careful optimisation of the following parameters: leaf strip size; incubation conditions; physical treatment; and enzyme concentration. The optimal parameters were combined in each individual step; the best enzyme concentration was determined to be 2% (*w*/*v*) cellulysin and 0.5% (*w*/*v*) macerase. Protoplast yield was found to be greatly affected by the enzyme concentrations. The isolated protoplasts were cultured at a density of 0.5 × 10^5^ to best sustain the highest cell division (33.2%) and a microcalli induction frequency of 17.9%. After 40 days of culture in a modified KM8P medium at 25 °C in darkness, visible microcalli were transferred to a solidified Murashige and Skoog (MS) medium with 1 mg L^−1^ 2,4-dichlorophenoxyacetic acid (2,4-D) for callus induction under a 16 h photoperiod. After 30 days of culture, compact organogenic calli were transferred into a solid MS medium with 6-benzylaminopurine (BA) alone or thidiazuron (TDZ) alone or in combination with BA or naphthalene acetic acid (NAA) for shoot regeneration. The maximum shoot regeneration frequency (63.3%) was observed in the medium with 1.5 mg L^−1^ TDZ alone. For the first time, a reliable protoplast isolation and regeneration system from mesophyll cells was established for *Duboisia* with high protoplast viability, successful microcalli formation, and intact plant regeneration. This innovation will significantly contribute towards the genetic enhancement of the *Duboisia* species.

## 1. Introduction

A plant cell devoid of a cell wall is termed protoplast and offers great potential for the genetic enhancement of many plant species via the fusion of such cells or by providing an easy platform for incorporating foreign DNA during genetic transformation [1]. Protoplasts also provide a one-of-a-kind possibility for medicinal species to be genetically enhanced. Bulk DNA transfer by protoplast fusion may result in novel germplasm with synergistic combinations of secondary metabolites or increased metabolites production [2]. Additionally, direct gene transfer through protoplasts is a rapid and efficient approach to introducing agronomically advantageous traits [3]. Recent scientific advances have made it possible to edit genes without chromosomal insertion by utilising clustered regularly interspaced short palindromic repeats (CRISPR)-mediated transient expression of RNPs and the regeneration of protoplasts [4]. This DNA-free genome editing system is ideal for generating foreign-DNA-free mutants delivering greater commercial value, unrestricted by current GMO legislations [5]. Despite the great potential and advantages of protoplast technology, regenerating plants from protoplasts is species-specific and remains the greatest obstacle to their widespread usage.

Establishing a reliable protoplast regeneration system is a prerequisite for the application of protoplast technologies, especially protoplast fusion and direct gene transfer [1]. Protoplast isolation is the first stage requiring an array of optimisations to achieve high-quality protoplasts with considerable yields. A protoplast can be isolated from different plant tissues and organs, such as leaf, suspension culture, hypocotyl, and cotyledon [6]. Protoplast isolation and regeneration success has been noted to be significantly influenced by both the choice of explant source and the specific genotype under consideration [7]. A suspension culture is widely used as a source material for protoplast isolation. However, the establishment of a suspension culture is time-consuming and associated with a high risk of somaclonal variation as well as the possibility of reducing the regeneration ability of cells [8]. Mesophyll cells are differentiated cells; thus, the direct use of mesophyll cells for protoplast isolation excludes many cycles of in vitro culture, reducing the risk of induced somaclonal variation [9]. Moreover, mesophyll protoplasts are the most commonly used protoplast for somatic hybridisation and efficient plant transformation, with simple and fast isolation procedures [10]. The vast majority of studies on mesophyll protoplast regeneration have been fruitful in herbaceous plants, such as rice [11], banana [12], and cabbage [13], whereas mesophyll protoplasts isolated from woody species have proven to be difficult to regenerate [14]. However, success in protoplast isolation and regeneration has been reported with several species, such as *Passiflora edulis* fv flavicarpa Degener. [15], *Malus pumila* [16], *Morus indica* [17], and *Phellodendron amurense* Rupr. [18].

The perennial shrub *Duboisia* (Solanaceae) is endemic to Australia [19]. This species contains abundant tropane alkaloids, scopolamine, and hyoscyamine, among other active ingredients [20]. Scopolamine, also identified as hyoscine, is the most commonly prescribed pharmaceutical for treating nausea, vomiting, motion sickness, and muscle spasms [21]. As the highest natural producer of scopolamine, *Duboisia* has been commercially cultivated for harvesting leaves for alkaloid extraction [22]. The growing demand for scopolamine exerts pressure on the *Duboisia* industry to provide consistent and sustainable raw scopolamine. The most sustainable way of increasing scopolamine production is through increasing the scopolamine content in leaves, which can be achieved through the development of new varieties via conventional breeding (taking over a decade to develop a variety) or through genetic manipulation with biotechnological tools. A simple and viable protoplast system would be highly beneficial to supporting future efforts in genetic enhancement to maximise scopolamine biosynthesis.

*Duboisia* protoplasts, with no exception, are recalcitrant to in vitro regeneration. Previous works on *Duboisia* protoplasts are only confined to cells from suspension cultures [23,24], where there is a high risk of induced somaclonal variation due to rapid cell division under high-hormone regimes. Mesophyll cells for protoplast preparation will alleviate the aforementioned issues; however, regenerating plants from mesophyll-derived protoplast is a formidable task. This study aimed to establish a reliable protoplast system using leaf tissues to avoid the high risk of mutation whilst achieving greater regeneration efficiency, building the foundation technology for future uses in *Duboisia* genetic enhancement approaches.

## 2. Results

### 2.1. Protoplast Isolation

#### 2.1.1. Optimisation of Protoplast Digestion

Through a series of step-by-step experiments, a reliable protocol for the isolation of high-quantity and -quality protoplasts from *Duboisia* mesophyll tissues was established. The protocol includes several optimisation steps that were conducted in a progressive manner.

##### Factor 1–Leaf Strip Size

The first optimisation step focused on the size of the leaf strips used in the protoplast digestion process. Two different sizes, 0.5–1 mm and 2 mm (Figure 1a), were compared for protoplast release. Based on the microscopic observation, more protoplasts were observed in the leaf strips of size 0.5–1 mm compared to 2 mm (Figure 1b,c).

##### Factor 2–Physical Treatment

The next optimisation step involved testing the effects of physical treatment, specifically vacuum infiltration, on protoplast digestion. The results showed that vacuum infiltration was highly effective for protoplast digestion. In comparison to the control group without vacuum treatment, the vacuum-treated leaf strips released more protoplasts, which were more uniformly round and exhibited a better quality (Figure 2a). In contrast, the control group released fewer protoplasts, which exhibited irregular shapes and a lower overall quantity (Figure 2b).

##### Factor 3–Incubation Conditions

Shaking the leaf strips on an orbital shaker at 50 rpm during incubation severely damaged the protoplasts, resulting in membrane ruptures and irregular shapes (Figure 3b). In contrast, static incubation (no shaking) resulted in uniformly round and healthy protoplasts (Figure 3a) and was used for subsequent protoplast isolation experiments.

##### Factor 4–Digestion Duration

The duration of enzyme digestion emerged as a crucial factor affecting protoplast release. It was clear that a 16 h incubation period under dark condition was the optimal duration for digestion. The results demonstrated that protoplast release was not observed before 10 h of digestion, and only a few protoplasts with uniform round shapes and a healthy appearance were released at 10 h of digestion (Figure 4a). In contrast, a marked increase in protoplast release was observed at 16 h of digestion, with a substantial release of protoplasts which were uniformly round and healthy (Figure 4b).

##### Factor 5–Enzyme Concentration

The final optimisation step for the protoplast isolation protocol involved determining the optimal enzyme concentration using the previously optimised parameters for leaf strip size, physical treatment, incubation condition, and digestion duration. Two different enzyme solutions containing varying concentrations of cellulysin and macerase were tested, and the released protoplasts were purified using gradient centrifugation (Figure 5a). Hemocytometry was used to determine the yield of purified protoplasts (Figure 5b), while viability was assessed using FDA staining (Figure 5c).

Protoplast yield was significantly affected by the concentration of cellulysin and macerase (Table 1). The combination of 1% (*w*/*v*) cellulysin and 0.25% (*w*/*v*) macerase could sustain no more than 1.9 × 10^5^ cells per gram of fresh weight. Doubling the enzyme concentration significantly raised *Duboisia* protoplast yield to 8.9 × 10^5^ cells per gram of fresh weight. As a result, a combination of 2% (*w*/*v*) cellulysin and 0.5% (*w*/*v*) macerase was determined as the optimum treatment and used for subsequent protoplast isolation procedures.

No statistical difference was observed in the protoplast viability between different enzyme treatments (Table 1). It was possible to improve protoplast viability to nearly 100% by reducing the enzyme digestion period. However, a shorter enzyme incubation period (10 h) dramatically diminished protoplast release (Figure 4a).

### 2.2. Plant Regeneration from Protoplasts

#### 2.2.1. Selection of Culture System and Basal Medium

In the preliminary experiment aimed at selecting a suitable culture system, five culture systems, including a liquid culture, a droplets culture, a solid–liquid double layer system, an agar pool culture, and an alginate bead culture, were compared (Figure 6). However, no protoplast development could be observed in any of the tested culture systems. This posed the question of the suitability of the culture medium for survival.

The liquid culture was then used as a standardised system to evaluate the effects on microcalli formation with different basal media, specifically B5, WPM, MS-NH_4_, and KM8P media (Figure 7). After 11 days of culture, it was found that only the KM8P medium was able to support protoplast development, as evidenced by enlarged protoplasts and protoplast division (Figure 7d). In contrast, the protoplasts cultured in other media failed to develop (Figure 7a–c).

#### 2.2.2. Microcalli Induction and Proliferation

Following the selection of the liquid culture system and KM8P medium as the suitable culture system and basal medium for *Duboisia* mesophyll protoplast development, an evaluation of plating density and hormone combinations was conducted to enhance protoplast division and microcalli development.

The plating density had a significant effect on culturing *Duboisia* mesophyll protoplasts. Table 2 depicts the effects of three protoplast plating densities on cell division after 11 days and microcalli induction after 40 days. The lower plating at 0.5 × 10^5^ resulted in the highest cell division efficiency at 33.2%, and a maximum of 17.9% microcalli induction frequency was observed in this case. A high protoplast density at 5 × 10^5^ diminished cell division and microcalli formation frequency substantially. Even at the density of 10^5^, nutritional competition resulting in necrosis and aberrant development was evident under a microscope when compared with the optimal density 0.5 × 10^5^ (Figure 8), despite no statistical difference being recorded on cell division and microcalli induction frequency. Therefore, the optimum plating density was determined to be 0.5 × 10^5^ and used for the subsequent regeneration process (Figure 9).

The induced microcalli with a diameter of 0.1–0.5 mm (Figure 9d) were transferred to a solid MS medium supplemented with different hormone combinations to evaluate their effects on the proliferation of microcalli into calli. After 30 days of culture, the hormone combination 1 mg L^−1^ 2,4-D + 0.5 mg L^−1^ BA supported the best microcalli proliferation at all three densities, producing a greater number of green and compact calli compared to the other hormone combinations. However, the calli produced from all the treatments exhibited a fuzzy surface texture, indicating anomalous cell proliferation (Figure 10).

A further investigation was carried out to examine the effects of BA on microcalli growth and address the fuzzy surface texture observed in the previous experiment. The microcalli induced at a plating density of 0.5 × 10^5^ were cultured on a solid MS medium supplemented with 1 mg L^−1^ 2,4-D with or without 0.5 mg L^−1^ BA (Figure 11). The deletion of 0.5 mg L^−1^ BA effectively eliminated the fuzzy surface texture from the induced calli, resulting in the production of green and healthy calli (Figure 11b).

#### 2.2.3. Plant Regeneration from Callus

To induce shoot regeneration, the healthy and compact calli were subcultured on an MS medium supplemented with various hormone combinations (Figure 12). After 6 weeks of culture, the calli treated with BA and free hormones exhibited a severe browning appearance, leading to mortality (Figure 12a–c). After 9 weeks of culture, shoot regeneration was observed on the calli treated with TDZ alone or in combination with BA or NAA. The shoot regeneration frequency was significantly influenced by the hormone treatments. The maximum shoot regeneration (63.3%) was observed on the calli treated with 1.5 mg L^−1^ TDZ alone (Figure 13a). Adding 1.5 mg L^−1^ BA or 0.1 mg L^−1^ NAA also stimulated shoot formation; however, significantly lower regeneration frequencies with delayed shoot growth were observed (Table 3). The use of free hormones medium and medium supplemented with BA alone or TDZ in combination with 0.5 mg L^−1^ NAA failed to induce shoots.

After shoot regeneration, these shoots rooted at a rate of 100% (Figure 13c). No morphological difference was observed between the protoplast-derived plants and the donor plants at the time of intact plant regeneration.

### 2.3. Growth Evaluation of Protoplast-Derived Plants

The growth performance of the protoplast-derived plants and their donor plants was further compared (Figure 14) through the assessment of plant height, stem girth, leaf number, canopy, and branch number over 3 months.

#### 2.3.1. Plant Height

No significant differences were observed in the data collected on plant height over the 12-week period between the protoplast-derived and the donor plants (Figure 15).

#### 2.3.2. Stem Girth

No statistically significant differences were observed in the stem girth between the protoplast-derived plants and the donor plants over the 12-week period (Figure 16).

#### 2.3.3. Number of Leaves and Canopy

Significant differences in the number of leaves and canopy were observed between the protoplast-derived plants and the donor plants. The data showed that, at week 6 and 7, the protoplast-derived plants produced significantly fewer leaves compared to the donor plants (Figure 17a). Similarly, from week 8 onwards, the canopy data indicated a significantly lower leaf yield in the protoplast-derived plants compared to the donor plants (Figure 17b).

#### 2.3.4. Number of Branches

The data collected on the number of branches showed a significant difference between the protoplast-derived plants and the donor plants from week 5 to week 12. The protoplast-derived plants exhibited fewer branches compared to the donor plants during this period (Figure 18).

## 3. Discussion

Protoplast technology has been deemed a significant and valuable technique for producing novel genotypes with desired features through protoplast fusion and genetic engineering. CRISPR/Cas9-mediated genome editing research is increasingly adopting protoplasts in the production of DNA-free mutants since this system has a greater potential for generating commercially feasible mutants with lower off-target effects than standard *Agrobacterium*-mediated transformation [25,26]. However, a protoplast-to-plant regeneration system is a prerequisite and remains challenging. A previous study successfully isolated and regenerated *Duboisia* protoplasts through a suspension culture system, but the fate of the isolated protoplasts did not move beyond transformation research [23]. In fact, genetic enhancement approaches generally require genetically stable materials, while the utilisation of suspension protoplasts carries the risk of somaclonal variation [27,28]. Mesophyll protoplasts, as a highly differentiated alternative, could limit this issue, but plant regeneration from mesophyll protoplasts is far more challenging. In this study, the first protocol of mesophyll protoplast isolation and regeneration for *Duboisia* has been established.

Several critical factors were optimised step-by-step for *Duboisia* mesophyll protoplast isolation, culture of protoplast, and intact plant regeneration. Starting from protoplast isolation, smaller leaf strips with a size of 0.5–1 mm resulted in a higher protoplast yield than the 2 mm leaf strips. This is likely due to the increased surface area to volume ratio of the smaller leaf strips, which may facilitate the diffusion of enzymes and aid in cell wall breakdown [29]. Relevant studies, such as the work by Liu et al. [30] and Kanai and Edwards [31], have also shown the importance of using a smaller leaf strip size for maximising protoplast yield in *Ricinus communis* L. and maize, respectively. Moreover, vacuum infiltration was found to be highly effective for *Duboisia* mesophyll protoplast isolation, resulting in more uniform and higher-quality protoplasts. The application of vacuum infiltration has been widely used to increase the efficiency and uniformity of protoplast isolation by aiding enzyme solution penetration into intercellular spaces, which is essential for the consistent release of intact protoplasts [32,33,34]. Shake incubation has been reported to be effective in speeding enzyme digestion, while a low shaking speed or static incubation can lead to longer incubation times and enzyme toxicity [33,35]. However, an excessive shaking speed may also lead to the disruption of chloroplast orientation and protoplast bursting [36]. In the present study, the shaking of leaf strips at 50 rpm during enzyme digestion resulted in the bursting of protoplasts, suggesting that shaking speed needs to be carefully optimised along with other factors, such as enzyme concentration and duration. A long duration (16 h) of static incubation, as demonstrated in this study, is a simple and effective method for achieving a sufficient yield of *Duboisia* mesophyll protoplasts. Exploring lower shaking speeds (below 50 rpm) could be beneficial to increasing protoplast yield; however, this may add-on further optimisation steps, with other digestion parameters. The balance of enzyme concentration and digestion duration is also essential for achieving high-quality protoplasts’ isolation. An excessive amount of enzymes can cause protoplast damage, whereas insufficient enzymes can result in incomplete digestion [37]. On the other hand, a prolonged digestion duration can lead to enzyme toxicity, while a shorter period can yield low-quality protoplasts [36]. For instance, Moon et al. [38] demonstrated that an 18 h enzyme incubation time yielded the highest extraction efficiency for potato mesophyll protoplasts, while a 5 h incubation led to low yields and a 24 h incubation resulted in decreased protoplast viability. The present study effectively demonstrated the balancing process for enzyme concentration and digestion duration through a series of progressive experiments: a final concentration of 2% (*w*/*v*) cellulysin and 0.5% (*w*/*v*) macerase with a 16 h digestion duration was found to be optimum for *Duboisia* mesophyll protoplast isolation.

An appropriate culture system and culture medium are essential for protoplast development, division, and regeneration. In contrast to the work by Kitamura et al. [23,24], which employed a B5 medium for *Duboisia* suspension culture protoplasts, the present investigation of mesophyll protoplasts found this medium to be ineffective. This highlights the notable influence of protoplast source on medium suitability. The present study also compared several more common media for protoplast culture and found that only the KM8P medium was able to support *Duboisia* protoplast development. This may be attributed to the presence of organic acids, casein hydrolysate and coconut water, which acted as necessary energy sources and provided vitamins, minerals, and sugars for protoplast development [6]. The superior performance of KM8P has been previously reported in the protoplast regeneration of *Gossypium klotzschianum* A. [39], *Dendrobium* [40], and *Diospyros kaki* L. [41]. Different culture systems can provide varying levels of nutrient availability and physical support [42], and the selection of a suitable system for successful protoplast development is species-specific and requires careful evaluation [43,44,45]. In this study, a liquid system along with the KMP8 medium was found to be a simple and effective method for protoplast development.

To further promote protoplast division and microcalli induction, plating density and hormone combination were also evaluated. It was found that protoplast culture and microcalli formation were negatively correlated with the tested plating densities. These data prove that protoplast culture densities were essential for cell division and microcalli induction in *Duboisia*. An excessive protoplast density in the culture medium depletes nutrients and causes phenol accumulation, resulting in a failure to maintain cell division and subsequent microcalli formation; this is well-demonstrated for the present study, as well as the works by Kang et al. [46] with *Petunia hybrida*, Avila-Peltroche et al. [47] with *Hecatonema terminale*, and Sangra et al. [36] with *Medicago sativa*. Moreover, the current findings revealed that protoplast plating density is highly dependent on the genotype and source of the protoplast at hand. Kitamura [23] reported that plating density at 5 × 10^5^ is optimum for culturing *Duboisia* suspension protoplasts. In the present study, a density at 5 × 10^5^ significantly reduced cell division and microcalli formation frequency for *Duboisia* mesophyll protoplasts. Furthermore, different combinations of 2,4-D and BA were evaluated on microcalli proliferation. In the literature, 2,4-D is recognised as a crucial plant growth regulator for microcalli development across various plant species, such as *Brassica* [48,49], *Camellia oleifera* [50], *Zoysia japonica* Steud. [51], and carrot [52]. Moreover, cytokinin or auxin are sometimes used in combination with 2,4-D to promote callus proliferation [17,53,54]. The present study demonstrated that the sole application of 1 mg L^−1^ 2,4-D is effective for *Duboisia* microcalli proliferation, as additional BA resulted in excessive cell proliferation leading to a fuzzy surface texture on the calli.

Plant growth regulators (PGRs) or plant hormones were found to be crucial for the shoot regeneration of mesophyll protoplast-derived calli in *Duboisia*. While Kitamura et al. [23,24] reported that the application of 5 mg L^−1^ BA successfully triggered shoot regeneration for suspension protoplast-derived calli, it was found to be ineffective for the present study. This, again, suggests that the PGR requirement for shoot regeneration is dependent on the source of the protoplast. The highest shoot regeneration was observed in media containing 1.5 mg L^−1^ TDZ alone or, to a lesser degree, with extra 1.5 mg L^−1^ BA or 0.1 mg L^−1^ NAA but not in media devoid of PGR or with other hormone combinations. These results highlight the superior effect of TDZ for the shoot regeneration of mesophyll protoplast-derived calli, which was also reported for apple [16], mulberry [17], *Tylophora indica* [55], and field cress [56]. In plant tissue cultures, the synthetic cytokinin-like phytohormone TDZ has been demonstrated to be the most effective agent for inducing shoot formation [57,58]. It has been reported that the cytokinin/auxin ratio is an essential factor in shoot regeneration [59]. The in vitro application of TDZ modulates endogenous hormone levels, particularly the cytokinin/auxin ratio [60]. It has been proven that the action mechanism of this PGR is tightly associated with the biosynthesis and transport of endogenous auxin as well as the accumulation of cytokinin [61,62]. The application of TDZ alone triggered effective shoot regeneration for the *Duboisia* mesophyll protoplast-derived calli. This indicates that additional NAA or BA might imbalance endogenous hormone levels, thus reducing shoot regeneration frequency.

A growth evaluation of the protoplast-derived plants was conducted, comparing several key agronomic traits with their donor plants. The growth evaluation experiment revealed that the protoplast-derived plants showed no differences to the donor plants when plant height and stem girth were compared. However, the protoplast-derived plants had fewer leaves and branches. This could be attributed to the enzymatic digestion, mechanical disruption, and osmotic stress caused during the protoplast isolation process [63]. However, it is not possible to compare the results as no similar studies are found in the literature. In the research sphere of tissue culture, some reports identify TDZ as a strong hormone known for promoting cell division and regeneration. However, it has tendency to induce somaclonal variations, particularly when used at high concentrations [64]. In a separate study, the application of a high level of TDZ (9.09 mg L^−1^) had a carry-over effect resulting in decreased shoot numbers and shoot lengths of micro-propagated Cymbidium [65]. Despite the observed variations in the leaf number and branch number of the protoplast-derived plants, the developed mesophyll protoplast isolation and regeneration system utilised highly differentiated leaves as the source material, thereby imposing a lesser risk of somaclonal variation. Further studies employing molecular and biochemical approaches such as amplified fragment length polymorphism (AFLP) analysis and sequence repeats (SSR) analysis can be used to identify whether the branching and leaf number differences are related to the genetic stability of protoplast-derived plants.

## 4. Materials and Methods

### 4.1. Protoplast Isolation

#### 4.1.1. Plant Materials and General Methods

A standardised method for protoplast isolation was developed with inputs from the related literature on several plant species [23,66,67]. Each digestion was carried out with 1 g of healthy and fully expanded leaves collected under aseptic conditions from in vitro cultures of *Duboisia* hybrid *D. myoporoides* × *D. hopwoodii* [68,69,70] with 10 mL of sterile enzyme solution. The enzyme solution was freshly prepared by dissolving digestion enzymes [Cellulysin^®^ (Sigma-Aldrich, Bayswater, VIC, Australia), Macerase^TM^ (Sigma-Aldrich, Bayswater, VIC, Australia)] in Solution A (0.4 M mannitol, 10 mM CaCl_2_, 20 mM KCl, 0.1% (*w*/*v*) BSA, and 20 mM MES, with a pH of 5.7, stored at −20 °C and thawed before use). Different enzyme concentrations were used in specific experiments during the optimisation process; thus, the details are included in the respective experiments. The enzyme solution was preheated at 60 °C for 10 min to inactivate the protease enzymes. Meanwhile, the leaves were shred vertically into strips using a sharp blade, taking much care not to bruise the tissues. The leaf strips were then immediately transferred into a 30 mm sterile Petri dish and flooded with freshly prepared filter-sterilised 10 mL of enzyme solution (refer to the experiments for the exact enzyme concentrations). The Petri dish containing the leaf strips and the enzyme solution was incubated under dark conditions at room temperature to achieve substantial protoplast release.

All the liquid media, buffers, and solutions utilised in this study were filter-sterilised using a 0.22 μm filter (Milles^®^-GS, Sigma-Aldrich, Bayswater, VIC, Australia), unless stated otherwise.

#### 4.1.2. Optimisation of Protoplast Digestion

The optimisation process considered several factors in the *Duboisia* mesophyll protoplast isolation process, including the following: leaf strip size, physical treatments, incubation conditions, digestion duration, and enzyme concentration. A series of progressive experiments were conducted in order to optimise one factor at a time to obtain a sufficient quantity of high-quality protoplasts required for further regeneration experiments. The outcomes of these experiments were visually assessed under a microscope; the slides for the microscope observations were prepared by randomly collecting three leaf strips from the digestion mixture. The image acquisition for the microscopic observations was taken consistently, capturing the area close to the edge of the selected leaf strip.

The optimisation procedure for an individual factor was not repeated; however, the optimal outcomes for each factor were amalgamated to form a final protocol and triplicated as independent experiments (Section 4.1.2, Factor 5) to ensure the robustness and reliability of the method.

##### Factor 1–Leaf Strip Size

Two different leaf strip sizes, 0.5–1 mm and 2 mm, were compared for protoplast digestion. The leaf strips were placed in a sterile Petri dish containing 10 mL of enzyme solution with 1% cellulysin and 0.25% macerase and cultured in darkness for 16 h at room temperature to observe protoplast release under a light microscope (Olympus BH2-RFCA, Notting Hill, VIC, Australia).

##### Factor 2–Physical Treatment

Following the optimisation of the leaf strip size, the effects of the physical treatment were evaluated. Specifically, vacuum infiltration was tested as a method to enhance protoplast release from the leaf tissue. The leaf strips were prepared as described previously and submerged in the enzyme solution. The aluminium foil-covered Petri dish containing the leaf strips and enzyme solution was then placed in a vacuum desiccator for vacuum infiltration at −40 kPa for 30 min. The Petri dish was subsequently incubated in darkness at room temperature for 16 h for the observation of protoplast release under a light microscope (Olympus BH2-RFCA).

##### Factor 3–Incubation Conditions

The next optimisation step was to evaluate the effects of incubation conditions on protoplast digestion. Similar to the previous steps, the leaf strips were prepared and treated with vacuum infiltration. Following this, the Petri dish was placed on an orbital shaker (Bioline, Narellan, NSW, Australia), shaking at 50 rpm, and incubated in darkness at room temperature for 16 h. Protoplast digestion was observed under a microscope (Zeiss Axio imager M2, North Ryde, NSW, Australia) to determine the optimal incubation conditions for protoplast release.

##### Factor 4–Digestion Duration

Subsequently, the optimisation was focused on the effects of digestion duration on protoplast release. After leaf preparation and vacuum infiltration, the leaf strips were incubated for different durations, including 3 h, 5 h, 10 h, and 16 h, at room temperature, in darkness, without shaking. Protoplast digestion was observed under a microscope (Zeiss Axio imager M2) after each time point to determine the optimal duration for protoplast release.

##### Factor 5–Enzyme Concentration

After these step-by-step optimisation processes, the optimal parameters determined were combined and used for the final optimisation step on enzyme concentration. The enzyme concentration was optimised by incubating the leaf strips in two enzyme solutions, one containing 1% cellulysin and 0.25% macerase and the other containing 2% cellulysin and 0.5% macerase. Data on protoplast yield and viability were recorded after purification. This experiment was triplicated as independent experiments to confirm the effectiveness of the developed protoplast digestion protocol.

#### 4.1.3. Protoplast Collection and Purification

The methods for protoplast collection and purification were modified from Zhang et al. [71] and Hu et al. [42]. Specifically, 5 mL of washing buffer composed of 154 mM NaCl, 125 mM CaCl_2_, 5 mM KCl, and 2 mM MES at pH 5.7 was added into the Petri dish to stop enzyme digestion. Subsequently, the Petri dish was gently agitated for 2–3 min to release the protoplasts. The protoplast suspension was filtered through a 40 μm cell strainer (Sigma-Aldrich) placed on a 50 mL sterile falcon tube, while the remaining leaf strips were gathered in a corner of the Petri dish and gently squeezed using a sterile pipette tip to release more protoplasts. Another 5 mL of washing buffer was added into the Petri dish to rinse the squeezed leaf strips; then, the contents were filtered through the strainer. The resulting protoplast suspension was centrifuged at 100× *g* at 4 °C for 5 min; the supernatant was carefully removed using a serological pipette. The protoplast pellet was washed three times with a washing buffer by centrifugation, following the same parameters. The washed pellet was resuspended in 2 mL of washing buffer, carefully overlayed on 6 mL of 18% (*w*/*v*) sucrose solution in a 15 mL falcon tube, and centrifuged at 80× *g* at 4 °C for 10 min with a swinging bucket centrifuge (Multifuge X4 Pro, Thermo Fisher, Scoresby, VIC, Australia). After the gradient centrifugation, a band of purified protoplast was collected at the interface between the sucrose and the washing buffer. The purified protoplasts were washed again with the washing buffer to remove any remaining sucrose solution.

#### 4.1.4. Protoplast Yield and Viability Test

A fluorescein diacetate (FDA) stock solution was prepared by dissolving 5 mg of FDA in 1 mL of acetone. An aliquot of 100 μL of protoplast suspension was mixed with 1 μL of FDA stock solution to make the final concentration of 0.01% in a 1.5 mL Enpendorf tube. The mixture was incubated at room temperature for 5 min in darkness. The protoplast yield and cell viability were determined using a haemocytometer under a fluorescence microscope (Zeiss Axio imager M2) at 420–490 nm. The viable cells stained by the FDA fluoresced of a yellow to green colour, while the dead cells remained unstained. Cell viability was calculated using the blow equation.
Cell viability (%) = Viable cells/Sum of viable cells and dead cells × 100% 

### 4.2. Plant Regeneration from Protoplasts

#### 4.2.1. Selection of Culture System and Basal Medium

To select the optimum basal medium and culture system for the isolated *Duboisia* mesophyll protoplasts, a preliminary experiment was carried out to determine the best culture system. In this experiment, the choice of the Gamborg (B5) medium as the basal medium [72], a hormone combination of 1 mg L^−1^ 2,4D and 1 mg L^−1^ zeatin, and a plating density of 5 × 10^5^ cells mL^−1^ were based on previous reports of *Duboisia* suspension protoplast cultures [23]. Five different culture systems were evaluated using the same basal medium, hormones, and plating density for comparison. The five culture systems evaluated were adapted from previously published methods, including liquid culture [73], droplets culture [23], solid–liquid double layer system [42], agar pool culture [42], and alginate bead culture [74]. However, no protoplast development was observed in any of the tested systems, leading to fine optimisation experiments for selecting suitable basal media.

To compare different basal media, the protoplasts were cultured using a liquid culture system. The same plating density (5 × 10^5^ cells mL^−1^) and hormone combination (1 mg L^−1^ 2,4D and 1 mg L^−1^ zeatin) were used for all the media tested, viz., B5, MS medium without ammonium (MS-NH_4_), woody plant medium (WPM) [75], and KM8P medium [76] (comprising PhytoTech Labs Inc. Kao and Michayluk modified basal salts, 0.58 g L^−1^ L-glutamine, 0.25 g L^−1^ Casamino acids, 0.25 g L^−1^ D-ribose, 72g L^−1^ glucose, and 20 mL L^−1^ coconut water). An aliquot of 1 mL of purified protoplasts at a 5 × 10^5^ cells mL^−1^ cell density was cultured in a 6-well plate (Sigma-Aldrich) and incubated at 25 °C in darkness. At 11 days of culture, protoplast development was assessed under an inverted microscope.

#### 4.2.2. Microcalli Induction and Proliferation

After the selection of the liquid culture system and KM8P medium as the appropriate culture system and basal medium for *Duboisia* mesophyll protoplast development, further evaluation was conducted to enhance protoplast division and microcalli development. Two crucial factors—plating density and hormone combinations—were evaluated.

Initially, the plating density was evaluated to determine the optimum density for microcalli induction. The purified protoplasts were adjusted to different densities (0.5 × 10^5^, 10^5^ and 5 × 10^5^ cells mL^−1^) in 1 mL of KM8P medium supplemented with 1 mg L^−1^ 2,4D and 1 mg L^−1^ zeatin and cultured in a 6-well plate (Sigma-Aldrich) at 25 °C in darkness. To promote cell division and microcalli formation, the dilution with 0.5 mL of half osmotic (36 g L^−1^ glucose) KM8P medium started 11 days after the initial plating, at weekly intervals. At 11 days of culture, the cell division frequency was computed as the percentage of protoplasts undergoing at least one cell division. After 40 days of culture, the frequency of microcalli induction was calculated as the percentage of protoplasts inducing visible microcalli. All the data were obtained under an inverted microscope (Olympus CKX41), with five random microscopic fields observed to calculate the average frequency for each treatment.

Subsequently, for each of the three plating densities, the induced microcalli with 0.1–0.5 mm were transferred to a solid MS medium [77] to evaluate different hormone combinations on microcalli proliferation. The MS medium was supplemented with 2% (*w*/*v*) sucrose, 0.7% (*w*/*v*) agar, and various combinations of 2,4D and BA, specifically, 1 mg L^−1^ 2,4-D + 0.5 mg L^−1^ BA, 2 mg L^−1^ 2,4-D + 0.5 mg L^−1^ BA, 1 mg L^−1^ 2,4-D + 1 mg L^−1^ BA, 0.5 mg L^−1^ 2,4-D + 1 mg L^−1^ BA, and 0.5 mg L^−1^ 2,4-D + 2 mg L^−1^ BA. The microcalli cultures were incubated at 25 °C with a 16 h/8 h (day/night) photoperiod (Lumilux cool daylight; OSRAM L 36W/865; PPFD 221.417 µmol m^−2^ s^−1^) for 30 days, after which the proliferated calli were visually assessed.

To investigate the impact of BA on microcalli proliferation and to address the observed anomalous cell proliferation leading to a fuzzy surface texture, an additional experiment was conducted. The microcalli were induced as previously mentioned with an initial plating density of 0.5 × 10^5^. The microcalli with a size 0.1–0.5 mm were cultured on a solid MS medium supplemented with 2% (*w*/*v*) sucrose, 0.7% (*w*/*v*) agar, and 1 mg L^−1^ 2,4-D, with or without 0.5 mg L^−1^ BA. After 30 days of culture under the same conditions described previously, the calli were evaluated visually to assess their proliferation and the impact of BA on microcalli growth.

#### 4.2.3. Plant Regeneration from Calli

Compact and well-organised calli induced from the protoplasts were transferred to a regeneration medium for shoot induction. To evaluate the effects of different hormones on shoot regeneration, the calli were transferred to an MS medium supplemented with BA (2 and 5 mg L^−1^) alone or TDZ (1.5 mg L^−1^) alone or in combination with BA (1.5 mg L^−1^) or NAA (0.1 and 0.5 mg L^−1^). The calli were subcultured for 3-week intervals until shoot regeneration. Data on shoot regeneration frequency were recorded after 9 weeks of culture. The regenerated shoots were subsequently rooted according to the procedure by Xue et al. [68].

### 4.3. Growth Evaluation of Protoplast-Derived Plants

The rooted plants were acclimatised according to Xue et al. [68]. After acclimatisation, the plants were re-potted in individual 125 mm Anova pots ^®^ with potting mixture UQ23 and maintained under open-field conditions at the University of Queensland for 3 months. The agronomy study of the protoplast-derived plants was carried out using a random block design, with two replicates consisting of ten protoplast-derived plants and ten donor plants. Data were collected each week on plant height, stem girth, number of leaves, and number of branches. To ensure accuracy in the data collection regarding leaf production, direct counting of the leaf number was limited to the initial 7 weeks. Subsequently, data on the canopy were obtained from the 8th week onwards as an indication of the leaf mass. The measurement of the canopy was taken by determining the widest distance from one branch tip to the opposite branch tip in two directions. The first measurement was taken at the point of maximum width of the canopy, while the second was taken perpendicularly to the first measurement, again, at the widest point. The canopy area was then calculated by multiplying the two measurements.

### 4.4. Data Analysis

All the data generated from this study were statistically analysed using the GraphPad Prism software (version 9.5.1, GraphPad Software Inc., San Diego, CA, USA). The data were analysed using independent samples *t*-test for two-group comparisons. For multiple group comparisons, an Anova test was conducted, and significant differences among the mean values were calculated using Tukey’s HSD test.

## 5. Conclusions

In this study, *Duboisia* mesophyll protoplast isolation as well as its plant regeneration protocols have been established for the first time. A number of factors—leaf strip size, physical treatments, incubation conditions, digestion duration, enzyme concentration, plating density, and plant growth regulator concentration and combination—were found to be very crucial for the success of *Duboisia* protoplast isolation and regeneration. The highest protoplast yield and viability could be achieved with the enzyme treatment of 2% (*w*/*v*) cellulysin and 0.5% (*w*/*v*) macerase, combined with a 30 min vacuum infiltration at −40 kPa, followed by a 16 h static incubation in dark conditions. The optimum plating density was determined to be 0.5 × 10^5^, supporting the highest cell division (33.2%) and microcalli induction (17.9%) frequency in a KM8P medium using a liquid culture system. After calli induction in a solid MS medium supplemented with 1 mg L^−1^ 2,4-D, the greatest shoot regeneration (63.3%) was observed with the medium containing 1.5 mg L^−1^ of TDZ alone. After this was followed by the rooting treatment, intact plant regeneration from *Duboisia* mesophyll protoplasts was achieved. The growth evaluation of the regenerated plants revealed no altered morphology, plant height, and stem girth compared to their donor plants; however, the protoplast-derived plants produced fewer leaves and branches. Despite these variations, the developed protoplast system, by employing highly differentiated mesophyll cells as the source material, imposed a lesser risk of somaclonal variation. This reliable mesophyll protoplast isolation and regeneration system could be further used as an effective tool for genetic manipulations including protoplast fusion and DNA-free genome editing.

## Figures and Tables

**Figure 1 plants-13-00040-f001:**
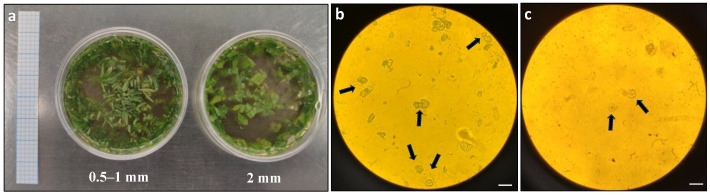
Comparison of protoplast release from *Duboisia* leaves with different strip sizes: (**a**) different size of leaf strips in enzyme solution; (**b**) microscopic observation of protoplast release from 0.5–1 mm leaf strips; and (**c**) microscopic observation of protoplast release from 2 mm leaf strips. The arrows indicate the observed protoplasts under an Olympus BH2-RFCA light microscope with 20× magnification. Graph paper grid size = 2 × 2 mm; scale bars = 50 μm (**b**,**c**).

**Figure 2 plants-13-00040-f002:**
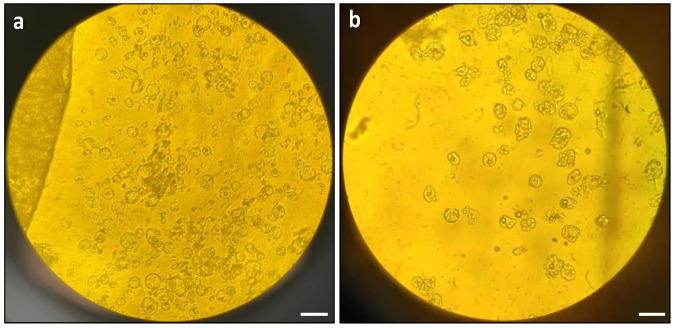
Effects of vacuum infiltration on protoplast digestion: (**a**) microscopic observation of protoplast release with vacuum infiltration; and (**b**) microscopic observation of protoplast release without vacuum infiltration. The images were taken with an Olympus BH2-RFCA light microscope with 20× magnification. Scale bars = 50 μm.

**Figure 3 plants-13-00040-f003:**
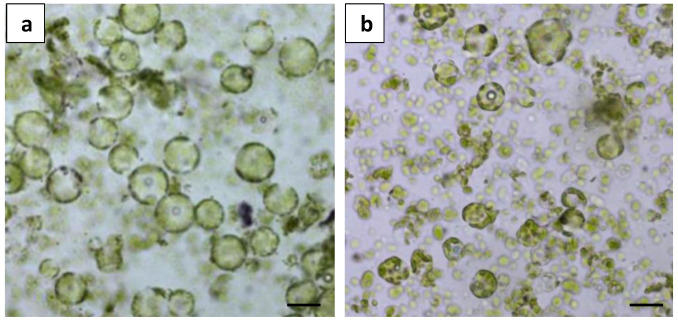
Comparison of different incubation conditions on protoplast digestion: (**a**) microscopic observation of protoplast release from static incubation; and (**b**) microscopic observation of protoplast release from shake incubation at 50 rpm. Scale bars = 50 μm.

**Figure 4 plants-13-00040-f004:**
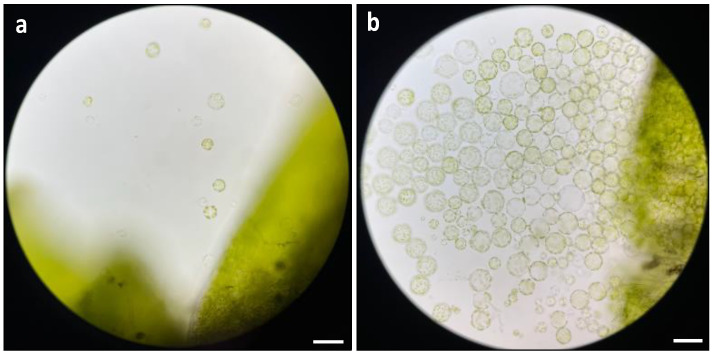
Comparison of different duration on protoplast digestion: (**a**) microscopic observation of protoplast release at 10 h of digestion; and (**b**) microscopic observation of protoplast release at 16 h of digestion. The images were taken with a Zeiss Axio imager M2 fluorescence microscope with 20× magnification. Scale bars = 50 μm.

**Figure 5 plants-13-00040-f005:**
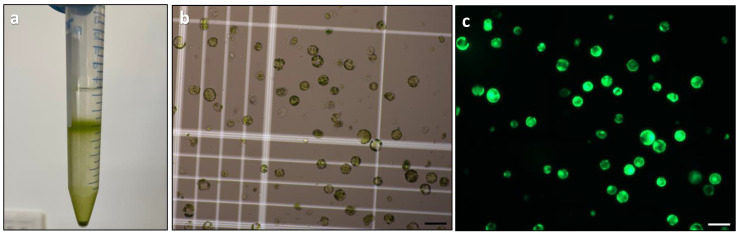
Protoplast purification and assessment of yield and viability: (**a**) protoplast purification using gradient centrifugation; (**b**) yield assessment of protoplasts using a haemocytometer; and (**c**) viability assessment of protoplasts by FDA staining. Scale bars = 100 μm (**b**,**c**).

**Figure 6 plants-13-00040-f006:**
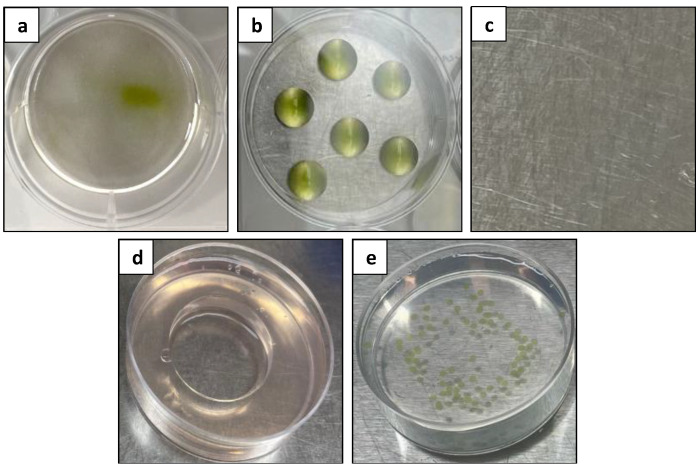
Comparison of different protoplast culture systems: (**a**) liquid culture; (**b**) droplets culture; (**c**) solid–liquid double layer system; (**d**) agar pool culture; and (**e**) alginate bead culture.

**Figure 7 plants-13-00040-f007:**
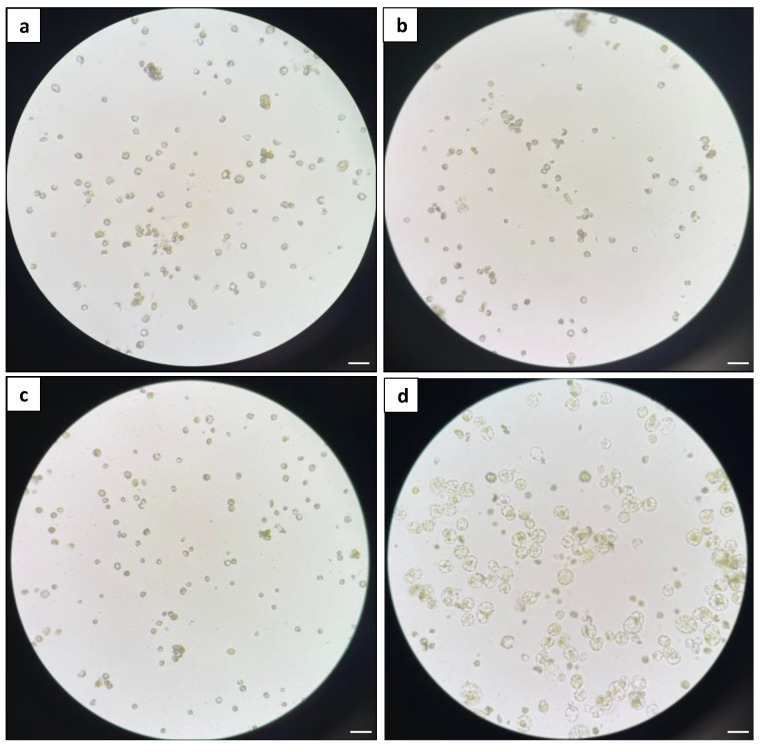
Comparison of different basal media: (**a**) microscopic observation of protoplasts in B5 medium; (**b**) microscopic observation of protoplasts in MS medium; (**c**) microscopic observation of protoplasts in WPM; and (**d**) microscopic observation of protoplasts in KM8P medium. The images were taken with an Olympus CKX41 inverted microscope with 20× magnification. Scale bars = 50 μm.

**Figure 8 plants-13-00040-f008:**
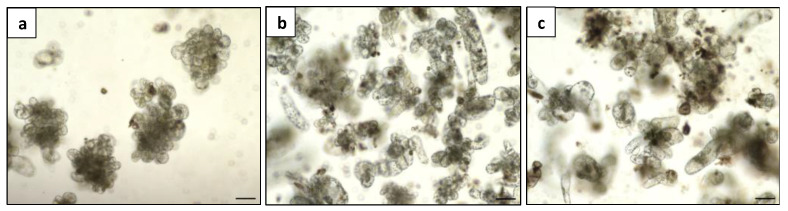
Microcalli induction of *Duboisia* mesophyll protoplast at three densities: (**a**) 0.5 × 10^5^; (**b**) 10^5^; and (**c**) 5 × 10^5^, bar = 100 μm.

**Figure 9 plants-13-00040-f009:**
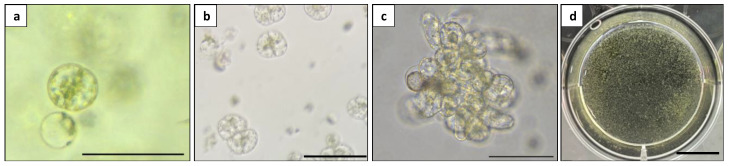
Microcalli induction from *Duboisia* mesophyll protoplast: (**a**) first cell division; (**b**) second cell division; (**c**) multiple cells stage; and (**d**) visible microcalli. Scale bars = 100 μm (**a**–**c**) and = 1 cm (**d**).

**Figure 10 plants-13-00040-f010:**
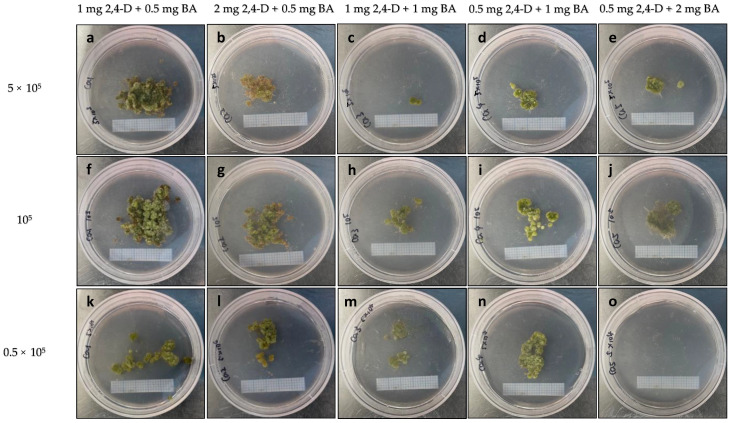
Microcalli proliferation on MS medium with the indicated hormone combinations (mg L^−1^) at different plating densities: (**a**–**e**) 0.5 × 10^5^; (**f**–**j**) 10^5^; and (**k**–**o**) 5 × 10^5^. Graph paper grid size = 2 × 2 mm.

**Figure 11 plants-13-00040-f011:**
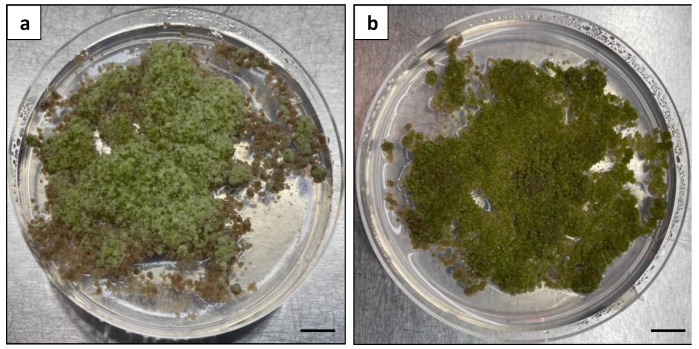
Effects of BA on microcalli proliferation: (**a**) calli with a fuzzy surface texture produced from the MS medium with 1 mg L^−1^ 2,4-D and 0.5 mg L^−1^ BA; (**b**) calli with a healthy green appearance produced from the MS medium with 1 mg L^−1^ 2,4-D alone. Scale bars = 1 cm.

**Figure 12 plants-13-00040-f012:**
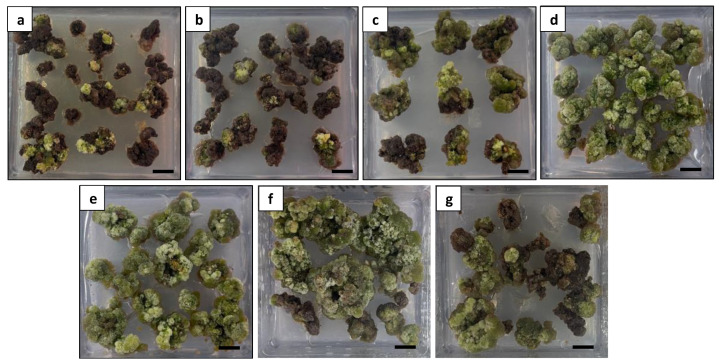
Comparison of different hormone combinations on callus appearance at 6 weeks of culture: (**a**) calli treated with free hormones exhibiting a severe browning appearance; (**b**) calli treated with 2 mg L^−1^ BA exhibiting a severe browning appearance; (**c**) calli treated with 5 mg L^−1^ BA exhibiting a severe browning appearance; (**d**) calli treated with 1.5 mg L^−1^ TDZ exhibiting a green and healthy appearance; (**e**) calli treated with 1.5 mg L^−1^ TDZ and 1.5 mg L^−1^ BA exhibiting a green and healthy appearance; (**f**) calli treated with 1.5 mg L^−1^ TDZ and 0.1 mg L^−1^ NAA exhibiting an overall healthy appearance with limited browning; and (**g**) calli treated with 1.5 mg L^−1^ TDZ and 0.5 mg L^−1^ NAA exhibiting a moderate browning appearance. Scale bars = 1 cm.

**Figure 13 plants-13-00040-f013:**
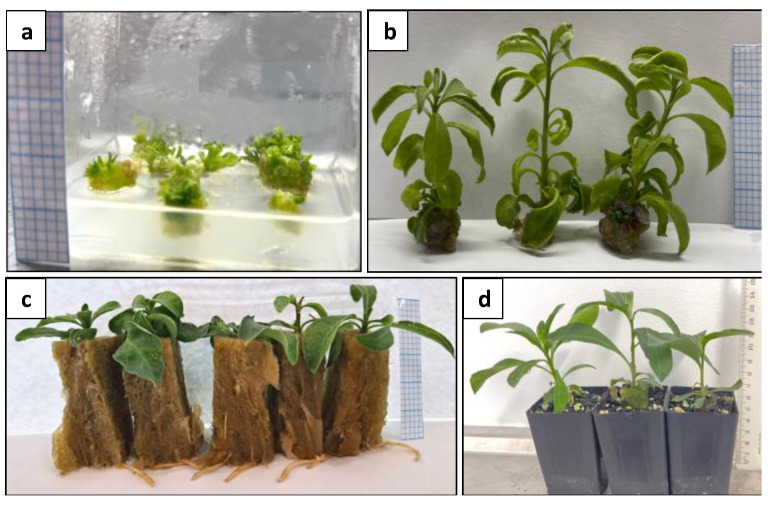
Plant regeneration from protoplast-derived calli: (**a**) shoot induction from calli; (**b**) regenerated shoots; (**c**) rooted shoots; and (**d**) acclimatised plants. Graph paper grid size = 2 × 2 mm (**a**–**c**); Scale bar = 1 cm (**d**).

**Figure 14 plants-13-00040-f014:**
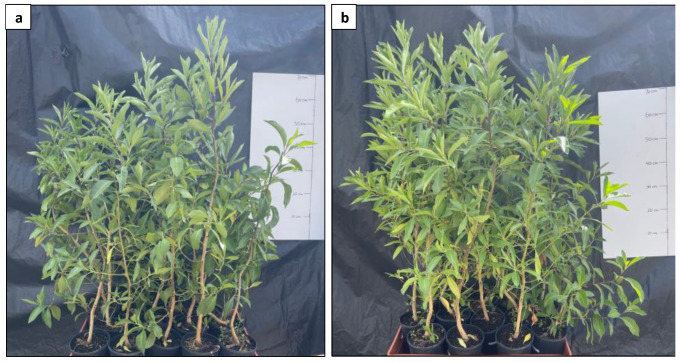
Comparison of growth performance between protoplast-derived plants and their donor plants. (**a**) protoplast-derived plants; and (**b**) donor plants.

**Figure 15 plants-13-00040-f015:**
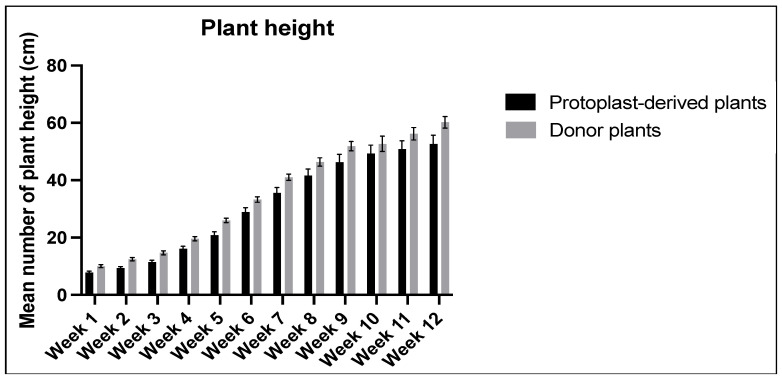
Comparison of plant height between the protoplast-derived plants and the donor plants over 12 weeks. All the data are presented as the mean ± standard error of the mean (SEM), *n* = 10, duplicated samples.

**Figure 16 plants-13-00040-f016:**
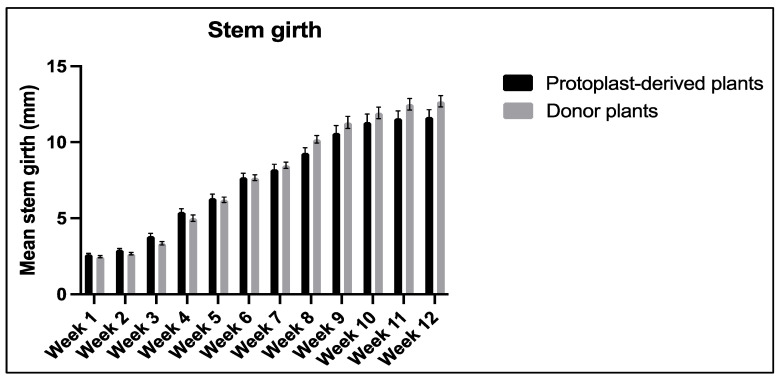
Comparison of stem girth between the protoplast-derived plants and the donor plants over 12 weeks. All the data are presented as the mean ± standard error of the mean (SEM), *n* = 10, duplicated samples.

**Figure 17 plants-13-00040-f017:**
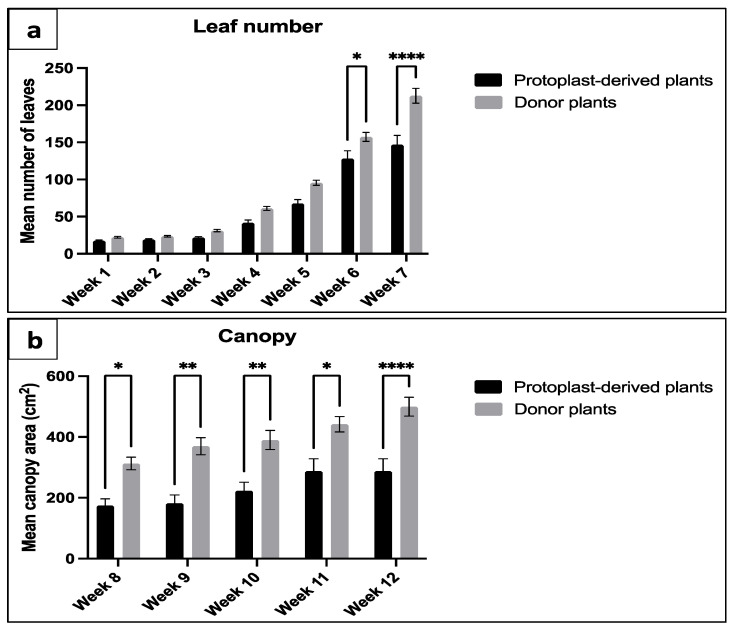
Comparison of leaf number and canopy between the protoplast-derived plants and the donor plants over 12 weeks: (**a**) comparison of leaf number from week 1 to 7; and (**b**) comparison of canopy area from week 8 to 12. All the data are presented as the mean ± standard error of the mean (SEM), *n* = 10, duplicated samples. Significant differences between the data sets are indicated as follows: * *p* < 0.05; ** *p* < 0.01; and **** *p* < 0.0001.

**Figure 18 plants-13-00040-f018:**
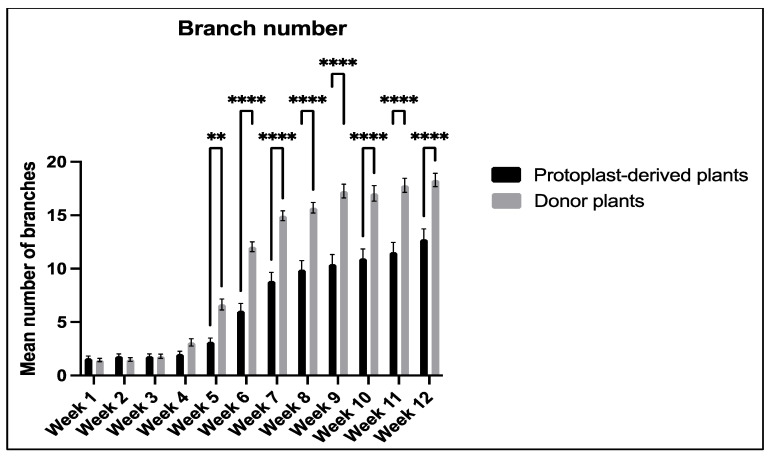
Comparison of branch number between the protoplast-derived plants and the donor plants over 12 weeks. All the data are presented as the mean ± standard error of the mean (SEM), *n* = 10, duplicated samples. Significant differences between the data sets are indicated as follows: ** *p* < 0.01; and **** *p* < 0.0001.

**Table 1 plants-13-00040-t001:** Effect of enzyme concentrations on *Duboisia* mesophyll protoplast isolation.

Cellulysin/Macerozyme Concentration (%, *w*/*v*)	Yield (×10^5^ Cells g^−1^ FW)	Viability (%)
1/0.25	1.5 ± 0.4 a	74.1 ± 6.5 a
2/0.5	8.9 ± 1.9 b	80.9 ± 8.9 a

Means and standard deviation within the same column followed by the same letter of the alphabet are not significantly different.

**Table 2 plants-13-00040-t002:** Effect of plating density on cell division and microcalli induction frequency of *Duboisia* mesophyll protoplast.

Protoplast Density (×10^5^ Cells g^−1^ FW)	Cell Division Frequency	Microcalli Induction Frequency
0.5	33.2 ± 6.3 a	17.9 ± 7.8 a
1	31.7 ± 5.9 a	14.8 ± 4.1 a
5	21.9 ± 2.8 b	5.9 ± 3.4 b

Means and standard deviation within a column followed by the same letter of the alphabet are not significantly different.

**Table 3 plants-13-00040-t003:** Effect of hormone type/concentration on shoot regeneration from mesophyll protoplast-derived calli of *Duboisia*.

Hormone Concentration (mg L^−1^)			Shoot Regeneration (%)
TDZ	BA	NAA	
0	0	0	0
0	2	0	0
0	5	0	0
1.5	0	0	63.3 ± 12.2 a
1.5	1.5	0	34.2 ± 8.0 b
1.5	0	0.1	27.9 ± 23.0 b
1.5	0	0.5	0

Means and standard deviation within the same column followed by the same letter of the alphabet are not significantly different.

## Data Availability

Data are contained within the article.

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
