# Peer review of "First Report on Mesophyll Protoplast Isolation and Regeneration System for the Duboisia Species"

_plants, 2023, doi:10.3390/plants13010040_

Round 1
Reviewer 1 Report
Comments and Suggestions for Authors
I think it is of great significance that we have completed a protoplast culture system for the medicinal plant Duboisia. However, as an academic paper, I judge that it is not complete enough to be accepted or published. Academic papers are supposed to be written accurately to ensure reproducibility, but even when it comes to cited references, there are many careless mistakes such as obvious mistakes, inconsistent titles, and not italicizing the scientific names of plants. It's too much. I hope that you will carefully rewrite your manuscript and resubmit it.
Author Response
Thank you for your comprehensive feedback on our manuscript. We appreciate the time you've dedicated to the review. Regarding your concerns, we understand the importance of accuracy and clarity in academic writing. We apologize for any oversights in the citation references, titles, and scientific names. We have carefully screened the manuscript and addressed these issues in our revised manuscript. We value your guidance and hope the corrections and changes address your concerns.
Reviewer 2 Report
Comments and Suggestions for Authors
Comments for the manuscript (plants-2713521) entitled “ First report on mesophyll protoplast isolation and regeneration system for Duboisia species”.
The submitted manuscript has some original and new information. The manuscript is clear and well presented. The language of the manuscript is good. The submitted manuscript is suitable for publication. However, a minor revision will improve the clarity of the manuscript. This manuscript may be accepted for publication after the completion of the minor corrections suggested below.
Title : I suggest the authors to change the title “ Efficient mesophyll based protoplast isolation and regeneration system for Duboisia species”.
Abstract: The abstract is clear and informative.
Line No. 12-14 : Remove the sentence “ Persistent efforts are underway to enhance the efficacy of the active ingredient scopolamine, employing both conventional breeding methods and advanced biotechnology tools”.
Key words : I suggest the following key words are more appropriate.
Duboisia spp., Corkwood, mesophyll protoplast isolation, regeneration of plants.
Introduction: The introduction is good and informative. The research problems explained were clear enough to reach the reader.
However, the authors failed to mention the earlier work in protoplast isolation. Include the following contribution in the Introduction.
Kitamura et al., (1989) Isolation and culture of protoplasts from cell suspension cultures of Duboisia myoporoides with subsequent plant regeneration.Plant Science. Vol: 60, Issue: 2, Page: 245-250.
Results: Results are clear and statistically interpreted.
The major problem is the Phrase- “(Error! Reference source not found)” appears throughout the result section. Rectify this problem.
The Table and Figure numbers should be mentioned in the appropriate places.
Line number 212 : What is in the brackets? 0.5 × 105 ( ??)
Discussion: Discussions are clear and good.
However, the authors have to compare the results of the earlier report Kitamura et al., (1989) Isolation and culture of protoplasts from cell suspension cultures of Duboisia myoporoides with subsequent plant regeneration.Plant Science. Vol: 60, Issue: 2, Page: 245-250.
Materials and Methods: The methodology followed in this research paper was appropriate.
What kind of explant source have been used? Did you extract the leaf from the in vitro raised plants? Or in vivo plants? - If so, give the explant sterilization methods.
Conclusion : Conclusions are adequate and acceptable.
Reviewer 3 Report
Comments and Suggestions for Authors
This manuscript (MS) mesophyll protoplast isolation and regeneration system for Duboisia species. The authors systematically optimized the parameters required for protoplast isolation and regeneration. Overall, MS is well-written and easy to read. The presentation of the results is adequate. I felt the discussion section needed to be concise for better quality.
Minor comments:
1. The figures 15-18 about the growth evaluation can be moved as supplemental figures.
2. It is not clear what is the optimum shaking condition for better protoplast yield.
3. The author mentioned that there is no altered morphology, but in another sentence, protoplast-derived plants produced fewer leaves and branches. It's contradictory and possible reasons to be explained.
Reviewer 4 Report
Comments and Suggestions for Authors
Paper "First report on mesophyll protoplast isolation and regeneration system for Duboisia species" presents a method for efficient isolation of protoplasts from this species.
Below the authors can find some suggestions for improving their manuscript.
Rows 55-56: ”Protoplast can be isolated from different plant tissues, such as leaf, suspension culture, hypocotyl and cotyledon" - please rephrase, leaf, hypocotyl and cotyledon are not plant tissues, they are organs.
Row 70: "passionfruit [15], apple [16], mulberry [17] and Phellodendron amurense rupr. [18].” – please provide the (correct) scientific names for all listed plant species.
Row 84:"Duboisia protoplasts with no exception to be recalcitrant to in vitro regeneration." - please rephrase, the verb is missing.
Rows 102, 104, etc: "(Error! Reference source not found.a)" - please check the whole text, many bibliographic references (or something else!) are missing.
Row 100” 2.1.1.1. Factor 1 - Leaf strip size" - I am not sure of the accuracy of the method used to estimate the number of protoplasts released using the 2 variants of leaf fragments of different sizes. In the image there are 2 photographed microscopic fields. Were several fields randomly measured to statistically ensure that one variant releases more protoplasts than the other?
Row 110: it is not sufficient to mention in the explanations at what magnification the photos were taken. Inserting a micrometric scale on the image is necessary. The same observation is valid for figures 2, 4, 7.
Rows 114-119: It is not clear if the results you refer to ("vacuum infiltration was highly effective for protoplast digestion.") are obtained in this experiment or are those of the authors indicated in the bibliographical references (which cannot be seen, due to the error loading). Or does that error refer to the figure number?? The observation from Row 100 is also valid here. How was the significant difference between the number of protoplasts obtained by the 2 methods assessed?
Rows 258-259: ”The shoot regeneration frequency significantly influenced by the hormone treatments." - to reformulate, the verb is missing.
All figures and tables must be cited in the text. Please insert specific references in the text related to the figures and photos that compose the figures when they are described in the text.
Row 433:" some reports identify as TDZ as a strong hormone" please reformulate.
Row 479: "under a light microscope" - the type of microscope used in the investigations must be mentioned in the Material and method chapter, not in the Results.
Row 492 "shaking at 50 rpm" - why didn't you try experimenting with other (lower) shaking speeds? It is possible that they had positive effects on the release of viable protoplasts.
Row 503: optimal
Row 508: "This experiment was triplicated" - why didn't you perform the other experiments in triplicate?
Row 536: below
Regarding the conclusions: do you think that the presented method can be extended to other species of Duboisia?
References still need to be corrected, there are still some mistakes, DOI is not provided for all papers, although it is available.
Comments on the Quality of English LanguageMinor editing of English language required.
Round 2
Reviewer 1 Report
Comments and Suggestions for Authors
I would like to correct just one point. Line 574: Light intensity is not specified in the lighting conditions. Should be described in PPFD or lx.
Author Response
The authors appreciate your careful observation and agree that light intensity should be specified in the methods for better accuracy and clarity. We have included the exact PPFD reading for the culture room used in this study. We hope our revised manuscript addresses your concerns.